

# Voluntarily controlled but not merely observed visual feedback affects postural sway

Shu Imaizumi[1,2], Tomohisa Asai[3], Kentaro Hiromitsu[4] and Hiroshi Imamizu[3,5]

[1] Graduate School of Arts and Sciences, The University of Tokyo, Tokyo, Japan
[2] Japan Society for the Promotion of Science, Tokyo, Japan
[3] Cognitive Mechanisms Laboratories, Advanced Telecommunications Research Institute International, Kyoto, Japan
[4] Graduate School of Letters, Chuo University, Tokyo, Japan
[5] Graduate School of Humanities and Sociology, The University of Tokyo, Tokyo, Japan

## ABSTRACT

Online stabilization of human standing posture utilizes multisensory afferences (e.g., vision). Whereas visual feedback of spontaneous postural sway can stabilize postural control especially when observers concentrate on their body and intend to minimize postural sway, the effect of intentional control of visual feedback on postural sway itself remains unclear. This study assessed quiet standing posture in healthy adults voluntarily controlling or merely observing visual feedback. The visual feedback (moving square) had either low or high gain and was either horizontally flipped or not. Participants in the voluntary-control group were instructed to minimize their postural sway while voluntarily controlling visual feedback, whereas those in the observation group were instructed to minimize their postural sway while merely observing visual feedback. As a result, magnified and flipped visual feedback increased postural sway only in the voluntary-control group. Furthermore, regardless of the instructions and feedback manipulations, the experienced sense of control over visual feedback positively correlated with the magnitude of postural sway. We suggest that voluntarily controlled, but not merely observed, visual feedback is incorporated into the feedback control system for posture and begins to affect postural sway.

## INTRODUCTION

Human body posture is stabilized by the feedforward and feedback control systems. In the feedforward control system, online comparison between predicted and actual body posture is made on the basis of a predictive signal computed by internal models (*Fitzpatrick, Burke & Gandevia, 1996*; *Van der Kooij et al., 1999*). In the feedback control system, concurrent multisensory afferences (i.e., visual, vestibular, and proprioceptive domains) are utilized for online maintenance of body part positions and balance (*Mergner & Rosemeier, 1998*; *Peterka, 2002*). Thus, for instance, unstable body posture during quiet standing can be observed in patients with vestibular disorders (*Dozza, Chiari & Horak, 2005*; *Fregly, 1974*) and in healthy individuals with transient proprioceptive deprivation

Corresponding author
Shu Imaizumi,
imaizumi@beck.c.u-tokyo.ac.jp,
shuimaizumi@gmail.com

due to ischemia (*Diener et al., 1984*). Furthermore, deprivation of visual input by closing the eyes robustly perturbs postural control (*Edwards, 1946*; *Lee & Lishman, 1975*; *Travis, 1945*). These findings suggest that unisensory information is crucial for intact postural control, even though other sensory modalities retain proper information for postural control.

## Postural sway modulated by visual feedback

The biofeedback technique, by which a quietly standing observer is exposed to additional unisensory stimulation interpreted from the online displacement of his or her center of pressure (CoP) on a force plate, has been utilized for training and rehabilitation for postural control (*Litvinenkova & Hlavacka, 1973*; *Takeya, Sugano & Ohno, 1976*; *Zijlstra et al., 2010*). For example, the auditory feedback technique, by which medio-lateral (ML) and antero-posterior (AP) displacements of observers' CoP are converted to a continuous tone of varying volume and pitch and delivered to the observers, has been reported to improve postural control in patients with vestibular disorders (*Dozza, Chiari & Horak, 2005*; *Dozza, Horak & Chiari, 2007*), whereas some studies have demonstrated the effectiveness of tactile feedback on the tongue (*Tyler, Danilov & Bach-y Rita, 2003*; *Vuillerme et al., 2007*).

The visual feedback technique, by which observers are presented with the online plot of their CoP displacement on a monitor in the coronal plane parallel to the observers' coronal, has been reported to decrease postural sway (*Gantchev, Draganova & Dunev, 1981*; *Litvinenkova & Hlavacka, 1973*; *Rougier, Farenc & Berger, 2004*; *Van Peppen et al., 2006*; *Zijlstra et al., 2010*). Literature suggests that there is a stabilizing effect of visual feedback on postural control in healthy adults, both young and old, and in patients with altered postural stability (*Dault et al., 2003*; *Freitas & Duarte, 2012*). There is still controversy regarding its effectiveness and feasibility for patients after stroke (*Geurts et al., 2005*; *Van Peppen et al., 2006*). The mechanism of postural stabilization by visual feedback has considered that the visual feedback provides additional visual inputs in order to integrate multisensory information for the purpose of stabilizing body posture during quiet standing. Some studies have demonstrated that magnification of visual feedback gain relative to actual CoP displacement can further help postural control, because when visual feedback gain is magnified, slight CoP displacements can be easily detected, facilitating the adjustment of postural control (*Cawsey et al., 2009*; *Jehu, Thibault & Lajoie, 2016*; *Rougier, Farenc & Berger, 2004*). Another factor of the biofeedback technique, spatiotemporal (in)congruence of visual feedback has also been studied. Visual feedback with a certain amount of delay (i.e., smaller than 900 ms) stabilizes postural control (*Rougier, 2004*), while larger delays can differentially affect low- and high-frequency fluctuations of CoP displacements (*Van den Heuvel et al., 2009*; *Yeh et al., 2010*). Horizontally-biased visual feedback requires horizontal compensatory postural adjustments, which can result in increased CoP displacements, but these displacements can be adapted after training (*Shiller et al., 2017*).

## Cognitive effects on postural control and their interactions with visual feedback

Postural control is also influenced by concurrent cognitive activities. Cognitive tasks performed during quiet standing, such as attentional or working memory tasks, affect postural control by reallocating resources for postural control and cognition. However, studies have reported mixed results, showing either increased, decreased, or unchanged postural sway (*Fraizer & Mitra, 2008*). Intentional effort to maintain posture has a key role in maintenance of postural control. Instruction to stand still (i.e., intention to minimize postural sway) has been consistently reported to stabilize postural sway, relative to the result of instruction to relax, although outcome postural indices differ among studies (*Loram, Kelly & Lakie, 2001*; *Mitra & Fraizer, 2004*; *Reynolds, 2010*; *Stoffregen et al., 2006*; *Ueta et al., 2015*; *Zok, Mazza & Cappozzo, 2008*). Instruction can even interfere with the effects of visual feedback on postural sway. For instance, healthy individuals using visual feedback have shown decreased postural sway when they are instructed to stand still, but when they are instructed to relax, they show postural sway comparable to that under non-feedback conditions (*Loram, Kelly & Lakie, 2001*). This finding suggests that visual feedback can be effective in maintaining postural control only when observers intend to minimize their postural sway.

However, it remains unclear whether intentional effort to utilize visual feedback to control posture affects postural sway or interacts with the effect of visual feedback itself. Several studies have already suggested that visuomotor coordination during walking (*Malone & Bastian, 2010*) and manual tasks (*Benson, Anguera & Seidler, 2011*) can be facilitated by instruction regarding explicit strategies for visual feedback. Given that the feedback system for postural control utilizes concurrent multisensory inputs, including vision, for online adjustment of body posture (*Mergner & Rosemeier, 1998*; *Peterka, 2002*), visual feedback might be particularly able to influence postural sway when observers have an explicitly-guided intention to control both their body posture and its visual feedback so as to accomplish a closed visuo-postural loop. In such a situation, consequently, amplitude and orientation of the visual feedback may be likely to modulate postural sway. Thus, it may be hypothesized that postural control with an intention to control visual feedback might be more influenced by visual feedback and its properties, such as feedback gain (*Cawsey et al., 2009*; *Jehu, Thibault & Lajoie, 2016*; *Rougier, Farenc & Berger, 2004*) and spatial orientation (*Shiller et al., 2017*), than it would be without such intention. Hence, there may be an interactive effect of intentional control and feedback manipulation on postural sway. This study sought to test this hypothesis.

## The present study

We examined whether and how intentional control of concurrent visual feedback of participants' postural sway affects their postural sway. In the present experiment, one group of healthy young adults was instructed to minimize postural sway while voluntarily controlling the concurrent visual feedback of their postural sway presented in a head-mounted display. The other group was instead instructed to merely observe the feedback and not intentionally use it for postural control. This experiment employed a
**Table 1 Characteristics of participants.** Mean value is followed by standard deviation in parentheses.

|  | Voluntary-control group | Observation group | Statistics for group differences |
|---|---|---|---|
| Sex | Male 7, female 3 | Male 5, female 5 | $\chi^2(1) = 0.83$, $p = .361$, $\varphi = .204$ |
| Age (year) | 19.40 (1.43) | 18.80 (0.63) | $t(12.39^*) = 1.21$, $p = .248$, $d = .543$ |
| Height (m) | 1.680 (0.091) | 1.671 (0.066) | $t(18) = 0.25$, $p = .803$, $d = .114$ |
| Weight (kg) | 55.20 (6.30) | 56.40 (7.29) | $t(18) = 0.39$, $p = .698$, $d = .177$ |
| Body mass index (kg/m$^2$) | 19.54 (1.41) | 20.12 (1.41) | $t(18) = 0.92$, $p = .369$, $d = .412$ |

Notes.
*Indicates Welch's correction for violation of the homogeneity assumption.

between-participants design to avoid potential carry-over effect and demand characteristics associated with the instructions. Moreover, to examine whether and how the instruction for intentional control enhances the effects of visual feedback properties (e.g., amplitude and orientation) on the postural sway, the visual feedback had two levels of gain and was with or without spatial incongruence (i.e., horizontal flip). We hypothesized that, in participants with explicitly-guided intentions to control visual feedback, high feedback gain would decrease (e.g., *Cawsey et al., 2009*) and spatial incongruence between visual feedback and CoP displacement would increase (*Shiller et al., 2017*) their postural sway.

## MATERIALS AND METHODS

### Participants

Twenty Japanese undergraduates aged 18–22 years participated in the present experiment for monetary compensation of 500 Japanese yen (approximately 4.5 US dollars). Their characteristics are summarized in Table 1. Half of the participants were pseudo-randomly assigned to the voluntary-control group, whereas the other half was assigned to the observation group. The two groups were comparable in sex and age. We also controlled their height (*Chiari, Rocchi & Cappello, 2002*), weight (*Hue et al., 2007*), and body mass index (*Greve et al., 2007*), each of which may affect postural control. All participants reported that they were right-handed without orthopedic conditions or a history of neurological or psychiatric disorders and had normal visual acuity with or without correction by contact lenses. They also had adequate sleep the night before the experiment. Written informed consent was obtained from each participant prior to the experiment. The present study was conducted in accordance with the Declaration of Helsinki and was approved by the local ethical committee of the Graduate School of Arts and Sciences, The University of Tokyo (approval number: 520).

Sample size was determined based on *a priori* power analysis using G*Power 3.1.9.3 (*Faul et al., 2007*) for an analysis of variance (ANOVA) of the within-between factors, because our main interest was the interactive effect of instruction (i.e., voluntary control, mere observation) on feedback manipulation. The power analysis indicated that at least eight participants for each of the two groups were required for a statistical power of .95, assuming a large effect size in ANOVA ($f = .40$: *Cohen, 1988*) and Type I error probability of .05.

## Apparatus

A force plate (Wii Balance Board; Nintendo, Kyoto, Japan) on a rigid and flat surface tracked the displacements of participants' CoP on the ML and AP axes with a sampling rate of 30 Hz. The CoP displacement data were collected and sent to a computer (R63/PS; Toshiba, Tokyo, Japan) via Bluetooth interface by an in-house custom program written in Hot Soup Processor 3.4 (ONION Software, Japan) using the open-source library WiiMoteLib 1.7 (http://wiimotelib.codeplex.com) running on Windows 7 Professional 64-bit (Microsoft, Redmond, WA, USA). The Wii Balance Board has been confirmed as a valid and reliable measurement of postural sway (*Clark et al., 2010*; *Clark et al., 2014*; *Imaizumi, Asai & Koyama, 2016*), while limitations of its measurement precision should also be noted (see 'Limitations').

Visual feedback of CoP displacement, instructions, and questions were presented on a head-mounted display weighing 330 g (HMZ-T2, Sony, Tokyo, Japan), which had an organic light-emitting diode display with a resolution of 1,280 × 720 pixels and a refresh rate of 60 Hz (*Hummel et al., 2016*). We used a head-mounted display in order to control viewing posture and distance, based on recent evidence suggesting that wearing a head-mounted display is unlikely to affect postural sway during quiet standing (*Morel et al., 2015*; *Robert, Ballaz & Lemay, 2016*) and that effects of instruction (*Mitra & Fraizer, 2004*) and visual motion perception (*Imaizumi et al., 2015*) on postural sway can be detected even when using such a display.

## Stimuli

Visual feedback of postural sway (i.e., CoP displacement) was displayed as a white square moving on a coronal plane parallel to the participants' coronal plane (Fig. 1). The square, which subtended at $1.0 \times 1.0°$ with a luminance of 28.40 cd/m$^2$, was presented centrally on a homogeneous black screen (0.40 cd/m$^2$) at the beginning of each trial. The screen subtended at $45.0 \times 24.7°$ with the same aspect ratio as surface of the force plate (432 × 237 mm). A 1-mm displacement of CoP on the force plate was synchronously transformed into 0.10° movement of the white square in the low gain condition and into 0.25° movement in the high gain condition. Anterior, posterior, leftward, and rightward displacements of CoP were translated into the upward, downward, leftward, and rightward movements of the square, respectively. We added the horizontally flipped condition, in which the leftward and rightward CoP displacements were translated into the *rightward* and *leftward* square movements, respectively. This flip was used to vary the effect of visual feedback on postural control (*Shiller et al., 2017*) and the subjective feeling of control over the moving square (*Asai & Tanno, 2007*; *Farrer et al., 2008*) by inserting spatial incongruence between bodily movement and visual feedback. In sum, there were four conditions of visual feedback: low gain, low gain flipped, high gain, and high gain flipped.

## Procedures

The experiment was conducted individually in a quiet, dimly lit room. After the briefing, participants removed their wrist and hand ornaments and shoes, put on the head-mounted display, and stood still on the horizontal center of the force plate with their hands down

**A** 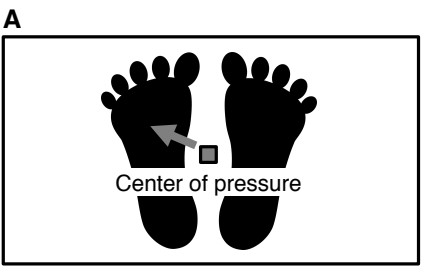   **B** 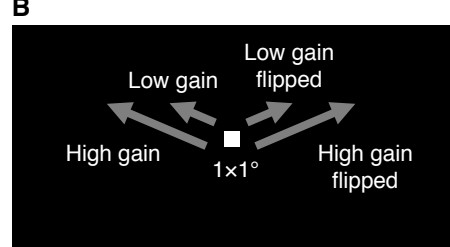

**Figure 1** **Schematic of the visual feedback of postural sway.** (A) The force plate tracked the displacement of participants' center of pressure. (B) The center of pressure displacement of 1 mm on the force plate corresponded to 0.10 and 0.25° displacement of a white square on the black screen in head-mounted display under the low and high gain conditions, respectively. For example, in the non-flipped and flipped conditions, when the center of pressure moved to the front left, the white square moved to the upper left and right, respectively.

at their sides and their heels together at a 30° angle between the medial sides of their feet (*Kapteyn et al., 1983*). Participants were asked to look straight ahead during the experiment.

In each trial, participants' CoP displacements were recorded for 31 s while being presented as a moving square on the display (i.e., visual feedback). Participants in both groups were instructed to concentrate on their postural sway and minimize it as much as possible (*Reynolds, 2010*). They were told that the moving square in the head-mounted display reflected their CoP displacement and postural sway. In the voluntary-control group, they were instructed to minimize their postural sway while voluntarily controlling and utilizing the moving square during the trial. In the observation group, they were instructed to minimize their postural sway while merely observing but not intentionally referring to the moving square. These instructions were presented on the display five seconds before each trial began, in order to inform participants of the task requirements and the onset of recording and stimulus presentation, and to prevent potential postural disturbance associated with stimulus onset and delayed stabilization. To check the validity of the instruction, immediately after each recording of postural sway, the display presented the following question: "To what extent did you feel that you were controlling the moving square?" with an 11-point Likert scale ranging from 0 (i.e., "Not at all") to 10 (i.e., "Extremely"). Participants' vocal responses were recorded by the experimenter. This question was adapted from the one used to measure sense of control over an external object (*Evans et al., 2015*; *Kalckert & Ehrsson, 2012*). Previous studies have used a similar index as that used by the present study to assess the sense of control and to reveal effects of between-participant factors (*Imaizumi, Asai & Koyama, 2016*; *Kokkinara et al., 2016*; *Peck et al., 2013*). Trials under each of four visual feedback conditions were repeated three times in a randomized order, for a total of 12 trials. The inter-trial intervals were 10 s each.

## Data analysis

Recorded CoP displacements during the first 1 s of all trials were excluded from analyses in order to eliminate potential outlying postural sway caused by stimulus onset and/or delayed stabilization. The data from the remaining 30 s were analyzed (*Kapteyn et al., 1983*). We calculated the total path length, ML path length, AP path length, and enveloped area of the CoP displacements. Total path length was calculated as the sum of the Euclidean distances between 900 successive data points (i.e., sampled at 30 Hz for 30 s). ML and AP path lengths were calculated as the sum of the ML and AP components, respectively, of the Euclidean distances between data points. Enveloped area was defined as the area enclosed by the outermost path of the CoP displacements. Indices of postural sway were computed using R 3.4.2 (*R Core Team, 2017*). We also used the *bivrp* package 1.0 (*Moral, Hinde & Demetrio, 2016*) to compute enveloped area.

## Statistical analysis

For each participant, each of the abovementioned subjective and postural indices was averaged for the three trials under each visual feedback condition. We first inputted the sense of control rating into a $2 \times 2 \times 2$ ANOVA with a between-factor (*Instruction*: voluntary control or observation) and two within-factors (*Gain*: low or high feedback gain; *Flip*: feedback without or with horizontal flip) in order to check the validity of the instruction. Subsequently, to test the effects of the instructed voluntary control of visual feedback on postural sway and the gain and spatial incongruence (i.e., flip) of visual feedback, we performed the same $2 \times 2 \times 2$ ANOVA on the total, ML, and AP path lengths, and enveloped area. As our interests were mainly in the main effects and interactions of Instruction, we performed post-hoc simple main effect analyses only when significant first- and second-order interactive effects of Instruction were found. Effect sizes in ANOVA were reported as generalized eta squared (*Olejnik & Algina, 2003*). Finally, to examine the relationship between the sense of control rating and postural sway in an exploratory manner, we computed Pearson's correlation coefficients between these indices from all participants under each of the four visual-feedback conditions (i.e., the degrees of freedom were 78). False discovery rate correction was applied for multiple comparisons (*Benjamini & Hochberg, 1995*). Significance level was set at $p < .05$. Hypothesis testing was conducted using SPSS 24.0 (IBM Corp., Armonk, New York) and R 3.4.2 (*R Core Team, 2017*).

# RESULTS

We performed ANOVA with a between-factor (Instruction) and within-factors (Gain, Flip) on the rating of sense of controlling visual feedback and postural measures. Main effects and interactions of these factors on each measure are summarized in Table 2.

## Sense of control rating: manipulation check

As expected, the voluntary-control group exhibited higher ratings for experienced sense of control over visual feedback than the observation group did, under all conditions (Fig. 2). This result was supported by a significant main effect of Instruction without any interactions; no effects were found for Gain or Flip (Table 2).

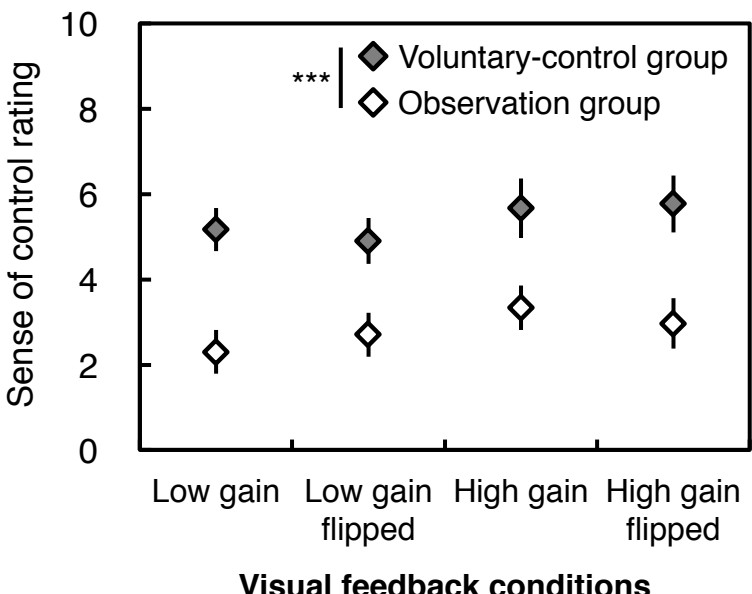

**Figure 2 Subjective rating of the sense of control over visual feedback of postural sway.** Error bars denote standard error of the mean. Asterisks indicate a significant difference between groups ($^{***}p < .001$).

**Table 2 Summary of the main effects and interactions of three factors on each dependent variable.** Degrees of freedom were 1 and 18. Statistically significant values ($p < .05$) are bolded.

|  |  | Instruction | Gain | Flip | Instruction × gain | Instruction × flip | Gain × flip | Instruction × gain × flip |
|---|---|---|---|---|---|---|---|---|
| Sense of control | $F$ | **16.46** | 2.06 | 0.06 | <0.01 | 0.14 | 0.37 | 2.95 |
|  | $p$ | **.001** | .169 | .804 | .972 | .710 | .552 | .103 |
|  | $\eta_G^2$ | **.355** | .036 | <.001 | <.001 | <.001 | .001 | .007 |
| Total path length | $F$ | 3.29 | **7.50** | 2.39 | **5.71** | 1.42 | **6.09** | **11.02** |
|  | $p$ | .086 | **.014** | .139 | **.028** | .249 | **.024** | **.004** |
|  | $\eta_G^2$ | .149 | **.007** | .002 | **.005** | .001 | **.003** | **.006** |
| ML path length | $F$ | **5.39** | 4.24 | 3.76 | 2.92 | 2.86 | 2.22 | **7.65** |
|  | $p$ | **.032** | .054 | .068 | .104 | .108 | .153 | **.013** |
|  | $\eta_G^2$ | **.214** | .012 | .006 | .008 | .004 | .001 | **.005** |
| AP path length | $F$ | 1.85 | **12.47** | 0.79 | **9.05** | 0.28 | **7.01** | **11.62** |
|  | $p$ | .191 | **.002** | .387 | **.008** | .605 | **.016** | **.003** |
|  | $\eta_G^2$ | .091 | **.005** | .001 | **.004** | <.001 | **.004** | **.006** |
| Enveloped area | $F$ | 2.93 | 0.34 | **12.04** | <0.01 | 3.86 | 4.09 | 0.82 |
|  | $p$ | .104 | .565 | **.003** | .959 | .065 | .058 | .377 |
|  | $\eta_G^2$ | .110 | .002 | **.054** | <.001 | .018 | .011 | .002 |

**Notes.**
ML, medio-lateral; AP, antero-posterior.
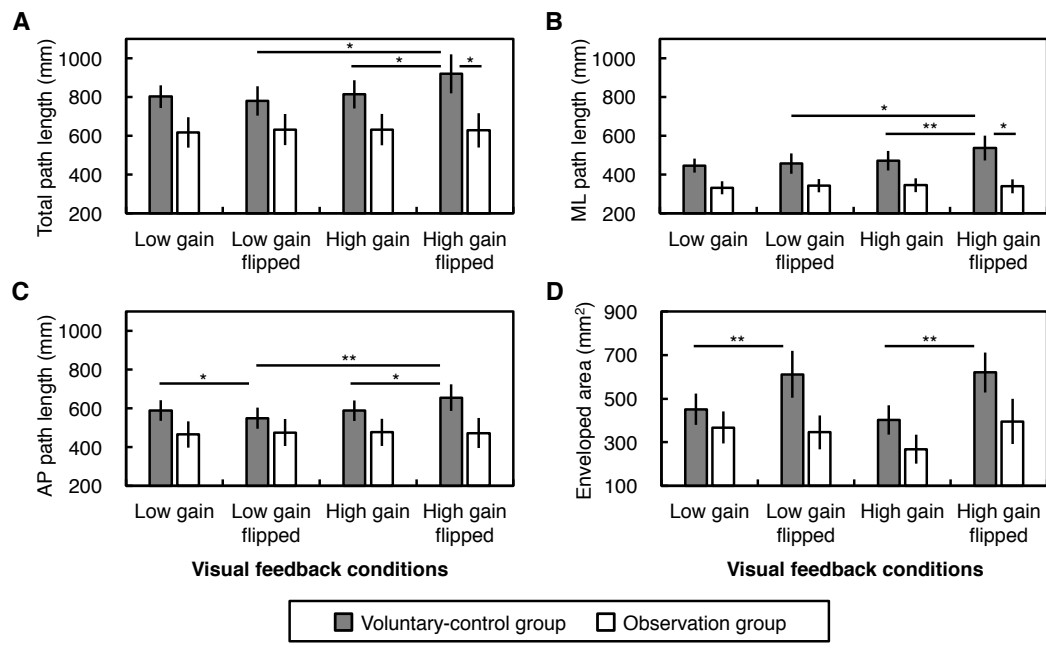

**Figure 3** **Magnitude of postural sway.** (A) Total path length, (B) medio-lateral (ML) path length, (C) antero-posterior (AP) path length, and (D) enveloped area of the center of pressure displacements. Error bars denote standard error of the mean. Asterisks indicate significant simple main effects ($^*p < .05$, $^{**}p < .01$).

## Magnitude of postural sway
### Path length
Results of the path lengths of CoP displacements are displayed in Figs. 3A–3C. We found a second-order Instruction × Gain × Flip interaction on the total path length, in addition to Gain × Flip and Instruction × Gain interactions (Table 2). Simple interaction analysis revealed that a Gain × Flip interaction was found in the voluntary-control group ($F(1,9) = 10.77$, $p = .010$, $\eta_G^2 = .018$) but not in the observation group ($F(1,9) = 0.81$, $p = .390$, $\eta_G^2 < .001$). Simple main effect analysis indicated that, in the voluntary-control group, greater total path length was found in the high gain flipped condition than in the low gain flipped and high gain non-flipped conditions ($F(1,9) = 10.17$, $p = .011$, $\eta_G^2 = .064$; $F(1,9) = 9.73$, $p = .012$, $\eta_G^2 = .039$, respectively). Furthermore, an Instruction × Gain interaction was found in the flipped condition ($F(1,18) = 9.18$, $p = .007$, $\eta_G^2 = .019$) but not in the non-flipped condition ($F(1,18) = 0.02$, $p = .900$, $\eta_G^2 < .001$), resulting in greater total path length under the high gain flipped condition in the voluntary-control group than in the observation group ($F(1,18) = 4.74$, $p = .043$, $\eta_G^2 = .209$).

A similar trend was observed for ML path length. There was a significant second-order Instruction × Gain × Flip interaction on ML path length (Table 2). Although no first-order interactions were observed, we performed an exploratory simple interaction analysis, revealing that a Gain × Flip interaction was found in the voluntary-control group ($F(1,9) = 5.24$, $p = .048$, $\eta_G^2 = .008$) but not in the observation group ($F(1,9) = 3.02$, $p =$

.116, $\eta_G^2 = .002$). An analysis of simple main effect indicated that in the voluntary-control group, greater ML path length was found for the high gain flipped condition than for the low gain flipped and the high gain non-flipped conditions ($F(1,9) = 6.38$, $p = .033$, $\eta_G^2 = .049$; $F(1,9) = 13.56$, $p = .005$, $\eta_G^2 = .034$, respectively). Moreover, an Instruction × Gain interaction was found in the flipped condition ($F(1,18) = 5.86$, $p = .026$, $\eta_G^2 = .020$) but not in the non-flipped condition ($F(1,18) = 0.21$, $p = .652$, $\eta_G^2 = .001$), resulting in greater ML path length under the high gain flipped condition in the voluntary-control group than in the observation group ($F(1,18) = 7.23$, $p = .015$, $\eta_G^2 = .287$).

As for AP path length, we found a significant Instruction × Gain × Flip second-order interaction in addition to Gain × Flip and Instruction × Gain interactions (Table 2). A simple Gain × Flip interaction was found in the voluntary-control group ($F(1,9) = 12.53$, $p = .006$, $\eta_G^2 = .023$) but not in the observation group ($F(1,9) = 0.54$, $p = .481$, $\eta_G^2 < .001$). Simple main effect analysis suggested that, in the voluntary-control group, AP path length was greater under the high gain flipped condition than under the low gain flipped and high gain non-flipped conditions ($F(1,9) = 17.03$, $p = .003$, $\eta_G^2 = .075$; $F(1,9) = 8.15$, $p = .019$, $\eta_G^2 = .033$, respectively), and smaller AP path length was observed for the low gain flipped condition than in the low gain non-flipped condition ($F(1,9) = 5.62$, $p = .042$, $\eta_G^2 = .015$).

Taken together, increased gain and spatial incongruence (i.e., flip) of the visual feedback lengthened ML and AP components of the CoP displacements only in the voluntary-control group, although the lengthening effect did not appear under some conditions.

*Enveloped area*
Results of the enveloped area of CoP displacements are displayed in Fig. 3D. We found no significant first- and second-order interactions (Table 2). However, given trends toward the significance of Instruction × Flip interaction ($p = .065$), we performed exploratory simple main effect analyses. As a result, there was a simple main effect of Flip in the voluntary-control group ($F(1,9) = 13.57$, $p = .005$, $\eta_G^2 = .118$) but not in the observation group ($F(1,9) = 1.24$, $p = .294$, $\eta_G^2 = .011$). These indicated that horizontal flip of visual feedback, but not feedback gain, increased the enveloped area of postural sway only in the voluntary-control group.

## Correlations among subjective and postural measures
Table 3 displays correlations between ratings of sense of control over visual feedback and magnitude of postural sway from both groups under each of the four feedback conditions. This analysis allowed us to check how these subjective and postural indices were correlated, regardless of experimental manipulations (i.e., instruction, feedback gain and flip). Results showed that sense of control rating positively correlated with total, ML, and AP path lengths. These results indicate that stronger sense of control over visual feedback is associated with the greater postural sway in path length. The enveloped area did not correlate with any measures.

**Table 3 Correlations among subjective and postural indices from all participants under each visual feedback condition.** Values are Pearson's correlation coefficients ($r$) followed by $p$ values (two-tailed; false discovery rate corrected) in parentheses. Degrees of freedom were 78. Statistically significant values ($p < .05$) are bolded.

|  | Sense of control | Total path length | ML path length | AP path length |
|---|---|---|---|---|
| Total path length | **.408 (.001)** |  |  |  |
| ML path length | **.492 (.001)** | **.900 (.001)** |  |  |
| AP path length | **.301 (.012)** | **.951 (.001)** | **.723 (.001)** |  |
| Enveloped area | .151 (.226) | .125 (.298) | .208 (.091) | .050 (.656) |

**Notes.**

ML, medio-lateral; AP, antero-posterior.

## DISCUSSION

The present study examined how intention to control visual feedback of postural sway and modification of visual feedback by gain magnification (low or high) and horizontal flip (with or without) have influences on postural sway. The intention to control was properly manipulated: participants in the voluntary-control group, who were instructed to minimize their postural sway while voluntarily controlling visual feedback, indeed rated their experienced sense of control over visual feedback more highly than did those in the observation group, who were instructed to minimize their postural sway while merely observing visual feedback without intentional reference to it for postural control.

### Voluntarily controlled, but not merely observed, visual feedback affects postural stability

We found that, as hypothesized, modification of visual feedback affected postural sway in the voluntary-control group and not in the observation group. Specifically, magnified gain and horizontal flip of the feedback increased path length of CoP displacements in ML and AP directions, whereas the enveloped area of postural sway was increased only by horizontal flip (see below for discussion regarding the difference between path length and area). Previous studies have demonstrated an interactive effect of intention to control *body posture* on the effect of visual feedback, indicating that visual feedback can affect postural stability only when observers are instructed to minimize postural sway (*Loram, Kelly & Lakie, 2001*). In contrast, the present results suggested an interactive effect of intention to control *visual feedback* on the effect of visual feedback itself, indicating that visual feedback can affect postural stability only when observers voluntarily control the visual feedback. In this situation, even artificially-added visual feedback should be incorporated into the sensorimotor loop in the feedback control system for online adjustments of body posture (*Mergner & Rosemeier, 1998*; *Peterka, 2002*). Although many researches have focused on the effects of additional sensory feedback on postural control (*Van Peppen et al., 2006*; *Zijlstra et al., 2010*), they might have overlooked how sensory feedback is voluntarily controlled and/or utilized by observers.

However, there seem to be two side effects of intentional control of visual feedback. First, the voluntary-control group appeared to show greater path lengths and enveloped area in all conditions than did the observation group, although a significant main effect of Instruction

was observed only for the ML path length. Explicitly-guided intention to minimize postural sway can robustly decrease postural sway more than just an intention to relax can (*Loram, Kelly & Lakie, 2001*; *Mitra & Fraizer, 2004*; *Reynolds, 2010*; *Stoffregen et al., 2006*; *Ueta et al., 2015*; *Zok, Mazza & Cappozzo, 2008*). Moreover, giving attentional focus to external objects while intending to minimize postural sway can also stabilize postural control (*McNevin & Wulf, 2002*; *Wulf et al., 2004*). Given that both groups in our experiment were instructed to minimize postural sway, and individuals in the voluntary-control group would have focused their attention on an external object (i.e., visual feedback), it would be plausible that the apparent differences in postural stability between groups resulted from the effect of intention to control the visual feedback per se. Second, contrary to our prediction, high gain feedback *increased* three types of path lengths (but only under the flipped conditions), while previous studies have suggested that high gain visual feedback *decreases* postural sway in healthy individuals (*Cawsey et al., 2009*; *Jehu, Thibault & Lajoie, 2016*; *Rougier, Farenc & Berger, 2004*). Possible explanations for the above side effects may be that, in order to voluntarily control and minimize the movement of visual feedback, participants were required to constantly adjust body posture against spontaneous and/or purposeful changes in the body reference configuration during quiet standing (*Danna-Dos-Santos et al., 2008*). Moreover, when feedback gain was magnified, participants had to adjust their body postures to a greater extent. Although there has been a controversy regarding the efficacy of visual feedback training on postural control (*Geurts et al., 2005*; *Van Peppen et al., 2006*), it might be speculated that the mixed outcomes of visual feedback training could be due to the lack of investigation on the influence of intentional effort to use visual feedback to adjust body posture.

Although intentional control of visual feedback may cause perturbing side effects, our correlation analysis indicated that there were positive correlations between sense of control ratings and total, ML, and AP path lengths across groups and conditions. This suggests that in order for additional visual feedback to affect postural control, the existence of both instruction to voluntarily control visual feedback and the experienced sense of control over the visual feedback are important, regardless of the magnification and spatial bias of the feedback. We should point out that even though the observation group was instructed not to intend to control the visual feedback, they did not indicate that they felt no sense of control (mean scores ranging approximately 2.0–3.5, see Fig. 2). It can be speculated that although the observation group did not have *a priori* intention to control visual feedback, they might have experienced a sense of control unconsciously generated from post-hoc inference (*Synofzik, Vosgerau & Voss, 2013*; *Wegner, 2003*), because they knew that the movement of visual feedback corresponded to their own CoP displacement. If so, in their violation of the instruction provided to the observation group, visual feedback might have had an influence on their postural control.

We should also clarify that the effects of gain magnification and horizontal flip on path length in the voluntary-control group were apparent only when both of them were applied, suggesting that each of these feedback modifications by itself was not strong enough to demonstrate an effect. While a previous finding suggests that horizontal flip of visual feedback could cause postural instability (*Shiller et al., 2017*), in our low gain conditions,

the flip did not result in any effect on total and ML path lengths, and even had a stabilizing effect on AP path length. This might be because participants may have had difficulty in detecting horizontal flip because of the low degree of feedback gain, and, consequently, the flip did not perturb their postural control. If this is the case, this explanation also accounts for the perturbation effect of horizontal flip in high gain conditions: participants could detect flip because of the large amount of visual feedback movement, and the flip thus affected their postural control. Nevertheless, horizontal flip increased enveloped area, regardless of feedback gain. This may highlight the lack of magnification of visual feedback in the high gain condition, and also suggest a potentially different nature of path length and enveloped area. Regarding sufficient amounts of feedback gain, relative difference between a high gain of 0.25° and a low gain of 0.10° corresponding to 1 mm CoP displacement might not be enough to increase postural sway, given that previous studies have reported that visual feedback with gains of 1.43° relative to 0.14° (*Rougier, 2005*) and 0.29° relative to 0.06° (*Jehu, Thibault & Lajoie, 2016*) decreased more postural sway (note that the authors transformed the original cm gain values into those of visual angles based on viewing distances reported in the cited papers). Further studies are needed to elucidate relationship between visual feedback modifications and intentional control of visual feedback, by applying wide-ranged, finely varied feedback gains and spatial rotations.

## Differences between sway path length and area

In our experiment, path lengths and the enveloped area of CoP displacements showed different tendencies in the effect of feedback modifications and different relationships with sense of control. The voluntary-control group exhibited total, ML, and AP path lengths subject to the effects of feedback gain and horizontal flip, and enveloped area affected only by horizontal flip. Furthermore, there was no correlation between path lengths and enveloped area. These results suggest a different nature of these indices, which may be interpreted by their different origins: sway path length, which reflects how frequently CoP fluctuates, originates mainly from the proprioceptive and motor systems (*Mauritz & Dietz, 1980*); in contrast, sway area, which reflects how widely CoP fluctuates, can originate from vestibular function (*Kapteyn & De Wit, 1972*), in addition to being influenced by the mechanical properties of body posture. We found that the three path lengths and sense of control rating correlated with each other, while the enveloped area did not correlate with any indices. These results not only further suggest the differences between sway path length and area, but may also indicate that sense of control contributes more to the motor-related sway component (i.e., path length) constituting a closed visuo-motor loop.

## Sense of control unaffected by visual feedback modifications

The subjective rating of the sense of control over visual feedback of one's own actions has been reported to be affected by intensity and spatial congruence of visual feedback. For example, faster movement of dots triggered by an observer's key press is likely to result in a stronger sense of control over the moving dots (*Kawabe, 2013*). Moreover, angular biases inserted into visual feedback of observers' manual actions using a joystick and computer mouse can reduce sense of control over the visual feedback (*Asai & Tanno,*

*2007*; *Farrer et al., 2008*). Contrary to our prediction made from these previous findings, the present results showed that sense of control over visual feedback of postural sway was not affected by feedback gain and spatial incongruence (i.e., flip). There were two potential explanations. First, the quantity of gain magnification was not enough to increase sense of control. Indeed, *Kawabe (2013)* reported that 8.5°/s movement of dots initiated by participants' key press induced stronger sense of control than did 2.1°/s movement, while 4.2°/s movement did not induce stronger sense of control than the 2.1°/s movement. Thus, our "high" feedback gain might indeed not be high enough to increase sense of control.

A second potential explanation, although speculative, is that the null effect on sense of control rating might be because of a potential difference between visuo-manual and visuo-postural relationships. Sense of control over external objects and sense of agency over one's own actions have been thought to stem from an internal forward model of the sensorimotor system in the brain (*Frith, Blakemore & Wolpert, 2000*), which includes the predictor and its comparator in order to match predicted and actual sensory feedbacks based on motor commands (*Wolpert, Ghahramani & Jordan, 1995*). Although many studies have experimentally manipulated spatiotemporal (in)congruence between sensory feedback and *manual* action and revealed the mechanisms of senses of control and agency (*David, Newen & Vogeley, 2008*; *Haggard, 2017*), little is known about the sense of control over sensory feedback of *full-body* movement such as postural control, except for locomotion (*Kannape & Blanke, 2013*; *Kannape et al., 2010*). Given that body posture is stabilized based not only on the predictive feedforward control system (*Fitzpatrick, Burke & Gandevia, 1996*), but also on the responsive feedback control system (*Peterka, 2002*), sense of control over sensory feedback of postural sway may arise in a manner different from that of manual action, whereby sense of control in postural control weighs its dependence less on the internal forward model than it does in manual action. Alternatively, we might assume that if the forward model is unlikely to predict single visual event (e.g., moving square) in the ecological environments as a consequence of postural sway and/or full-body movement, other than optic flow (*Fajen, 2007*), sense of control would not be affected by (in)congruence of visual feedback, regardless of gain and flip.

### Limitations

We have three limitations of note. First, although validity of the Wii Balance Board for postural measurement has been confirmed (e.g., *Clark et al., 2010*), recent studies have suggested its limited measurement precision, such as inconsistent sampling rate and poor signal to noise ratio (*Clark et al., 2018*; *Leach et al., 2014*). Our results might have been affected because we sampled the data using a constant rate within our custom program but did not calibrate the force parameter from the Wii Balance Board. Second, because we gave instructions to manipulate participants' intention to control the visual feedback of postural sway and assessed the experienced sense of control by subjective rating, we cannot exclude a possibility that the sense of control rating might have been biased by demand characteristics. Considering that participants' responses did not approach extreme values and varied across individuals (i.e., the voluntary-control group scored approximately 4–6, while the observation group scored approximately 2–4, Fig. 2), it may be plausible that

the group difference in the responses were not biased and caused by the instructions. Nevertheless, to address this potential issue, objective measures could be helpful. For example, intention to control and/or experienced sense of control may be reflected in the time-series regularity of postural fluctuations, such as fractality and self-affinity (*Delignieres et al., 2003*), as the degree of cognitive involvement in postural control (e.g., attention) can indeed be reflected in the postural time-series regularities (*Donker et al., 2007*; *Stins et al., 2009*). Finally, we employed instructions to and not to intentionally control visual feedback as a between-participants factor, in order to avoid potential carry-over effects and demand characteristics, which might affect the results. However, we cannot exclude the potential confounding effects of any individual differences in postural control and susceptibility to instruction. Further investigation to overcome the above limitations would facilitate deeper understanding of the effect of voluntary control over visual feedback on postural control.

## CONCLUSIONS

This study suggested that the observation of magnified and horizontally flipped visual feedback of postural sway in a quiet standing position can affect postural sway, but only when individuals intend to control the movement of visual feedback; moreover, the experienced sense of control correlates with postural sway. Voluntarily controlled, but not merely observed, visual feedback may be incorporated into the feedback postural-control system and affect postural sway.

Our findings contribute to an increased understanding of the potential role of intention and mental set in postural biofeedback techniques for healthy and impaired individuals. Although speculative, mixed outcomes from trials using postural biofeedback (*Geurts et al., 2005*; *Van Peppen et al., 2006*) might have resulted from a lack of consideration for the role of intentional control over the biofeedback. Thus, one direction for future clinical and therapeutic studies may include development of a new training protocol. It may be fruitful to further investigate how intentional control of sensory feedback and an experienced sense of control can influence postural control in patients who have experienced a stroke (*Shumway-Cook, Anson & Haller, 1988*), have a vestibular disorder (*Fregly, 1974*), or whose postural stability is likely to be perturbed. An additional future direction, from the perspective of psychology and psychiatry, would be to elucidate the relationships between sense of control or agency and whole-body movement (i.e., postural control and gait), particularly in schizophrenic and schizotypal individuals, who tend to experience a weakened sense of control and impaired self-other discrimination (*Asai, 2016*; *Franck et al., 2001*), in addition to healthy individuals.

## ACKNOWLEDGEMENTS

We would like to thank Ryota Hatakama for his help during data collection and three reviewers for their helpful comments on a previous version of the manuscript. The pilot experiment for this study was conducted in the NTT Communication Science Laboratories where SI and TA had formerly worked.

### Funding

This work was supported by Grant-in-Aids for JSPS Research Fellow (16J00411) and Young Scientists (B) (17K12701) to Shu Imaizumi, and a Grant-in-Aid for Scientific Research on Innovative Areas "Understanding brain plasticity on body representations to promote their adaptive functions" (26120002) to Hiroshi Imamizu from the Japan Society for the Promotion of Science, and partially supported by the Commissioned Research of the National Institute of Information and Communications Technology "Research and development of technology for enhancing functional recovery of elderly and disabled people based on non-invasive brain imaging and robotic assistive devices". There was no additional external funding received for this study. The funders had no role in study design, data collection and analysis, decision to publish, or preparation of the manuscript.

### Grant Disclosures

The following grant information was disclosed by the authors:
Grant-in-Aid for JSPS Research Fellow: 16J00411.
Grant-in-Aid for Young Scientists (B): 17K12701.
Grant-in-Aid for Scientific Research on Innovative Areas: 26120002.
Commissioned Research of the National Institute of Information and Communications Technology.

### Competing Interests

The authors declare there are no competing interests.

### Author Contributions

- Shu Imaizumi conceived and designed the experiments, performed the experiments, analyzed the data, prepared figures and/or tables, authored or reviewed drafts of the paper, approved the final draft.
- Tomohisa Asai and Hiroshi Imamizu conceived and designed the experiments, authored or reviewed drafts of the paper, approved the final draft.
- Kentaro Hiromitsu analyzed the data, authored or reviewed drafts of the paper, approved the final draft.

### Human Ethics

The following information was supplied relating to ethical approvals (i.e., approving body and any reference numbers):

The present study was conducted in accordance with the Declaration of Helsinki and was approved by the local ethical committee of the Graduate School of Arts and Sciences, The University of Tokyo (approval number: 520).

### Data Availability

Raw data can be found in the Supplemental Information.

## Supplemental Information

Supplemental information for this article can be found online at http://dx.doi.org/10.7717/peerj.4643#supplemental-information.

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
