# Peer review of "Voluntarily controlled but not merely observed visual feedback affects postural sway"

_PeerJ, doi:10.7717/peerj.4643_

## Round 0.1 · original submission · Major Revisions

Although the reviewers generally commented positively on the manuscript, they raised several concerns that need to be addressed in the revised manuscript. In particular, they commented on the presentation and interpretation of DFA analysis. Please also add the diffusion plots and/or power spectra from which the scaling exponents were estimated to Fig. 4. In addition, reviewers commented on the limitations of the Wii balance board and the sense of control assessment.

Reviewer 1 ·

Basic reporting

Overall this manuscript is clear and the topics is worth investigating, and the hypotheses are supported by previous literature. My only suggestion would be to have an native English speaker proofread the manuscript before re-submission. Further comments are addressed in the attached pdf.

Experimental design

Overall the experimental protocol is well-described and adequate to test the hypotheses. My major concern relates to the use and interpretation of DFA. My detailed comments can be found in the attached pdf, but I want to highlight the principal comment here: the authors suggest that the lower DFA values observed in the voluntary-control group reflect 'more' self-similarity, which is not the case (deviations toward 0.5 reflect less persistence).

While the rest of the study is very clear and rigorous, it seems that the application of DFA is lacking rigor and a clear rationale. As mentioned in the attached pdf, I suggest the authors to either remove completely DFA analyses (which may be necessary due to the very low sampling rate, 30 Hz; see Rhea et al., 2015), or to dig deeper in the analysis and interpretation of the results (e.g., DFA of CoP velocity, taking into account the cross-overs, etc.).

Validity of the findings

With the exception of DFA results, the results seems valid and tend to support the hypotheses.

Annotated reviews are not available for download in order to protect the identity of reviewers who chose to remain anonymous.

Reviewer 2 ·

Basic reporting

Imaizumi and colleagues investigated how the intentional control of visual feedback of postural sway influences the balance control of healthy adult participants, during quiet standing. They found that visual feedback increased the postural sway of individuals who voluntarily controlled the visual feedback. They suggested that the visual system is integrated into the postural control system by the closed visuo-postural loop mechanism.
The study is clearly and adequately written. It contains a nice historical review of studies of visual feedback and its effect on postural sway. While the study provides new knowledge about the effect of visual sensory information in control of posture during quiet standing, there are some issues that require the authors' attention.

Lines 65-66: be more specific in this sentence “…effectiveness and feasibility for patients”. What kind of patients you referee to?

Experimental design

The authors should state more clearly the possible contribution, implications or relevance of the present study to the field of research.

Line 142: the authors state that the participants had normal visual acuity. Did you make any test? How was it verified?

Besides the Wii balance board has been considered to be “…a valid and reliable measurement of postural sway….” as stated in the text, it is known that this device has some limitations (inconsistent sample rate, COP signal error and calibration method). Did the authors calibrate the equipment? What calibration method was used? How did you solve the question regarding to sample rate? Maybe re-sampling the raw data to ensure a consistent sample rate might help with this issue.
I suggest that such questions and limitations should be clearly presented in the text, the methods section needs some improvement.

Validity of the findings

The results are clearly described and alternative interpretations are discussed.

Lines 380-383: I suggest revising this sentence. The larger postural sway does not necessarily can be interpreted as a postural instability. It should take in account that central, peripheral and mechanical factors lead to spontaneous changes in the body reference configuration, the individuals make constant postural corrections during quiet upright standing posture.

Lines 472-473: the authors should be cautious when using a specific interpretation: “….sway area, which reflects how widely CoP fluctuates, originates from vestibular functions.” The amplitude of COP displacement is not necessarily originated from vestibular function, but also may reflect the mechanical properties of the body required for postural stability during upright standing.

Additional comments

The manuscript was carefully written and I appreciate the effort to collect such a data.

The study provides novel results about the effect of visual sensory information in control of posture during quiet standing, however there are some issues that require the authors' attention.

·

Basic reporting

The present manuscript has clear and professional English, achieving the journal standards regarding structure, figure quality and raw data supplied.

Concerning the INSTRODUCTION, it is clear and well-structured. The introduction is full of references about all it has been done in postural control and the effect of visual feedback. However, I miss an explanation about the relevance of the main aim of the study. In addition, I have some questions that are indicated in the reviewed PDF (please, check the comments). The main problem of this experiment is in the EXPERIMENTAL DESIGN, please, check the specific comments in the experimental design section and the comments in the reviewed PDF. The RESULTS and DISCUSSION are influenced by the design so I will wait to review them when the issues will be addressed, although, the authors can see some specific comments in the reviewed PDF.

Experimental design

In my humble opinion, the design of the experiment is not the most appropriate to address the goal of the study, it has some scarcities. The manipulation of the level of intention to control the visual feedback through the given instruction has some issues:

1. The answer of the participants to the question "to what extend did you feel that you were controlling the moving square? could be influenced by the instruction the authors gave to the participants. In other words, if the researcher said to the participant: do not pay attention to the moving square, and 30 s later the researcher asked him/her: did you pay attention to the moving square?... What do you think the answer is going to be? That could have happened or not but it is something out of the control of the researchers.

2. Despite the previous issue, the "sense of control" is a subjective variable that has been used in previous studies to assess the feel of controlling the movement. However, those studies, it has been used to compare the level of intention within one only group. It is said, they compared the level of controlling the movement of the participants in one situation with the level of controlling the movement of the same participants in another different situation. However, in this study, the authors have compared the rating of this variable between different groups giving different instructions. How could the authors know that the differences between groups are caused by the instruction if they do not know if both groups exhibit differences under the same situation with the same instruction?

3. The same problem exist with the rest of the variables, not just the sense of control. How do we know that the differences found are caused by the manipulation of the introduction, gain or flipped condition?

From my point of view, this big problem needs to be addressed. I suggest the following idea: using the same 2 groups, assess their postural control (all the variables used in the study) and the sense of control under the same exact situation, with the same instruction. You could use the situation where you ask them to pay attention to the moving square or the one where they should not pay attention to it, even both.
In that way, the authors will be able to check if the groups are or not different in all the used variables in the study. Thus, if they are not different the results of the study can be related to the given instruction, the gain or the horizontal flipped condition.

In addition, a better explanation about why the gain and the flipped condition are included in the study is needed.

Regarding the description of the method, the ethical standard and so on, I believe the authors did a great job. However, a better definition of the research question and to fix the big problem in the design of the protocol need to be addressed.

Validity of the findings

The design of the experiment needs to be improved to be able to evaluate the validity of the findings.

Additional comments

The main idea of the paper is interesting and the authors could find relevant discoveries for postural control research but several aspects of the paper need to be improved, mainly the design of the protocol.

Apart from these comments, please, check the specific comments in the reviewed PDF. I hope you find my suggestions useful.

---

## Round 0.2 · Minor Revisions

The reviewers raised some additional points that need to be addressed before the manuscript can be accepted.

Reviewer 1 ·

Basic reporting

The manuscript quality has improved, in particular the English.

Experimental design

The research design is thoroughly described and seems very rigorous.

Validity of the findings

The findings supports the conclusions. Limitations are identified, and future perspectives and proposed.

Additional comments

The authors have made significant efforts to improve the quality of their manuscript. I believe they addressed each and every comments from all reviewers.

·

Basic reporting

1. The authors have addressed the issues in the introduction.

2. L 113 and L312. Personally, I think the use of the DFA as an index of intentionality could have made the paper stronger. As I specified in the previous revision, it has been used in several studies to report the level of attention or intentionality the participants have (Stins et al. (2009) Gait Posture; Donker et al. (2007) Exp Brain Res). I hope you take it into account for future studies. The variable you use to talk about the voluntary control of the movement is subjective and it could have been supported by the DFA results.

3. L 136. I am so sorry but I still do not found the reference you made in the Procedure section regarding the pseudo-randomly assignment of the groups. Please, could you point it out? I do not think is a very relevant information to report but if you refer to it in the method it should be there.

Experimental design

1. Regarding the bias of the way the authors assessed the sense of control, the fact that the voluntary-control group scored approximately 4–6 on the sense of control rating scale across four feedback conditions, while the observation group scored approximately 2–4 can be caused precisely by the question you asked and the given instruction. Even if the groups were not biased toward an extreme response, as you say, this variable has its limitations so I would suggest adding it in the Limitation section.

2. I agree on the use of a between-participants design and the reasons the authors gave to support their decision. However, that does not avoid the issue with the “sense of control” variable that has been used. Since the authors talk about several studies which have measured the effects of between-participants factors by employing a similar index to assess the sense of control I would suggest referring some of them? Note: Asai & Tanno, (2007). The Relationship Between the Sense of Self-Agency and Schizotypal Personality Traits. Journal of Motor Behavior, 39, 3; used within-participant multivariate analysis of variance.

Validity of the findings

No comments

Additional comments

Thank you very much for addressed all the comments I made in the previous review. I hope you found them useful.

---

## Round 0.3 · accepted · Accept

The authors have adequately addressed the final comments.

#